# Preliminary Assessment of the Protective and Antitumor Effects of Several Phytoene-Containing Bacterial and Microalgal Extracts in Colorectal Cancer

**DOI:** 10.3390/molecules29215003

**Published:** 2024-10-22

**Authors:** Gloria Perazzoli, Cristina Luque, Antonio León-Vaz, Patricia Gómez-Villegas, Rocío Rengel, Ana Molina-Márquez, Ángeles Morón-Ortiz, Paula Mapelli-Brahm, José Prados, Consolación Melguizo, Antonio Meléndez-Martínez, Rosa León

**Affiliations:** 1Institute of Biopathology and Regenerative Medicine (IBIMER), Biomedical Research Center (CIBM), 18100 Granada, Spain; gperazzoli@ugr.es (G.P.); cristinaluque@ugr.es (C.L.); jcprados@ugr.es (J.P.); melguizo@ugr.es (C.M.); 2Instituto de Investigación Biosanitaria de Granada, Ibs. GRANADA, 18012 Granada, Spain; 3Department of Anatomy and Embryology, University of Granada, 18016 Granada, Spain; 4Laboratory of Biochemistry, Center for Natural Resources, Health and Environment (RENSMA), University of Huelva, Avda. de las Fuerzas Armadas s/n, 21071 Huelva, Spain; antonio.leon@dqcm.uhu.es (A.L.-V.); patricia.gomez@dqcm.uhu.es (P.G.-V.); anamariamolinamarquez@gmail.com (A.M.-M.); 5Food Color and Quality Laboratory, Facultad de Farmacia, Universidad de Sevilla, 41012 Sevilla, Spain; amortiz@us.es (Á.M.-O.); pmapelli@us.es (P.M.-B.); ajmelendez@us.es (A.M.-M.)

**Keywords:** antitumor extracts, colorless carotenoids, colorectal carcinoma, healthy diets, microalgae, phytoene

## Abstract

The identification of new functional food constituents is a priority to improve the prognosis and prevention of colorectal cancer (CRC). In this study, several bacterial and algal phytoene-enriched extracts were obtained, and their potential activity against oxidative damage and their ability to inhibit proliferation and cell migration in several human colon-adenocarcinoma-derived cell lines were assessed. The main conclusions indicate that total extracts of *Sphingomonas echinoides* and *Chlorella sorokiniana* exhibited the highest protective effect against oxidative damage. All extracts enhanced the activity of detoxifying enzymes, particularly importantly the increase of NAD(P)H:quinone oxidoreductase activity, which reached a value 40% higher than that of untreated control cells upon exposure to *Escherichia coli* extracts. *Staphylococcus haemolyticus* and transgenic *E. coli* extracts significantly arrested the migration capacity of both cell lines, while S. *haemolyticus* and *C. sorokiniana* extracts inhibited cell proliferation by 15 to 20% compared to untreated cells. These results point to these extracts as potential antioxidant complements able to protect cells against oxidative damage and with a moderate ability to inhibit the proliferation and migration of CRC tumor cells, paving the way to design functional foods or probiotic formulations with preventive properties against oxidative stress-related diseases, such as cancer, or as starting point for purifying anticancer compounds.

## 1. Introduction

The current treatment of CRC (colorectal cancer), the third most common type of cancer and second leading cause of cancer-related mortality, is mainly based on surgery or systemic chemotherapy and/or radiotherapy treatments. However, chemotherapeutic agents (i.e., 5-fluorouracil, capecitabine, oxaliplatin, irinotecan, etc.) show serious limitations, including a large number of adverse effects that exert a substantial impact on the general well-being of the patient, which may require the interruption of treatment [1]. Taking into account the high frequency of this type of cancer, its high mortality, and the foreseeable increase in its incidence over the years [2], the development of new therapeutic strategies and the identification of new functional nutrient components is essential for prevention and for improving the prognosis of patients with CRC.

Fruits and vegetables contain bioactive compounds with antioxidant, anti-inflammatory, and anti-aging properties which have been demonstrated to enhance the effect of antitumor classical drugs [3,4]. Among these bioactive phytochemicals, carotenoids, versatile isoprenoid compounds biosynthesized by all higher plants and algae, as well as by some bacteria, archaea, fungi, and arthropods, have received considerable attention. The importance of carotenoids in food science and nutrition has been based for many years in their role as natural colorants and their ability to combat vitamin A deficiency, as some of them are precursors of this essential nutrient. However, a growing body of evidence indicates that carotenoids can benefit human health in several ways [5]. Their dietary intake has been related to a reduced incidence of several chronic diseases, such as obesity, diabetes, cardiovascular disease, neurodegenerative disorders, macular degeneration, and some types of cancer. Mechanistically, the primary benefits of carotenoids can be explained by their antioxidant potential, either interacting directly with reactive oxygen species or through the induction of the expression of antioxidant enzymes. However, carotenoids may also provide health benefits through their conversion into vitamin A, absorption of light, modulation of gene expression, modulation of communication between cells, or interaction with the microbiota, among others [6,7].

Carotenoids are also very important in feeds for aquatic and terrestrial animals which, like human beings, are not able to synthesize carotenoids de novo but need to acquire them from the diet [8]. There are around one thousand cataloged carotenoids, of which only very few are colorless. The two main dietary colorless carotenoids are phytoene and phytofluene, which are the precursors of virtually all carotenoids. Although much attention has been paid to colored carotenoids, uncolored ones have been largely neglected in studies related to food science and technology, nutrition, or health. However, it is now clear that they are present in commonly consumed foods, are readily bioavailable, and can be involved in health-promoting biological actions and even provide aesthetic benefits [9]. Phytoene and phytofluene are present in food staples, such as tomatoes, carrots, apricots, or citrus, among others, and contribute significantly to total carotenoid intake, as demonstrated in two countries with different dietary patterns: Luxembourg and Spain. Particularly, phytoene was the second most consumed carotenoid in both countries [10,11].

In addition to traditional plant foods, microorganisms of different origins can be alternative sources of colorless carotenoids and other secondary metabolites. Along with their higher growth rate and productivity compared to plants, their easier metabolic modulation, their ability to grow in harsh conditions, and their contribution to CO_2_ sequestration and wastewater bioremediation [12], some are part of the human and animal microbiota and can synthesize essential compounds with health-promoting effects [13].

Despite their already commented on importance in the diet, there are still few studies addressing the potential health benefits of colorless carotenoids. This is especially surprising as one of the earliest works reporting the possible antitumor activity of carotenoids featured phytoene, which was reported to retard the appearance of UV-B-induced skin tumors and reduce their number in a rodent model [14]. Possible anti-prostate cancer activities of phytoene have also been reported in rodents [15]. Regarding absorbing UV light, there are several studies suggesting that the colorless carotenoids phytoene and phytofluene, on their own or from tomato products, can protect against light-induced damage, as shown in guinea pigs [16] and humans [17,18]. More recently, there has been evidence that phytoene, phytofluene, and lycopene can protect against nicotine-induced oxidative stress-mediated pancreatic islet dysfunction in rats. The nicotine-induced changes in fasting blood glucose, oral glucose tolerance, and glycated hemoglobin levels were reduced by the intake of the individual carotenoids and a combination of them [19]. On the other hand, phytoene and phytofluene have been shown to account for ~25% of the carotenoids in human adipose tissue, and their content in such tissue is inversely correlated with diastolic blood pressure [20]. All these precedents suggest that new uncommon colorless carotenoid sources can provide new therapeutic adjuvant therapies and new ingredients of sustainable healthy diets for CRC prevention.

Although phytoene is usually present in small amounts as a biosynthetic intermediate, its accumulation has been reported to be induced in higher plants [21] and microalgae [12,22] by treatment with bleaching herbicides, such as norflurazon, which causes inhibition of the enzyme phytoene desaturase by competition with its cofactors [23]. On the other hand, although high phytoene-producing bacteria are mostly unknown, the bacterial species *Sphingomonas echinoides* and *Staphylococcus haemolyticus* have been described as able to accumulate phytoene. *S. echinoides* (strain B-3127) forms colorless colonies and is a rarity within species of the *Sphingomonas* genus, which are typically pigmented by colored carotenoids [24]. *Staphylococcus haemolyticus* is a coagulase-negative *Staphylococcus* species (CoNS), found for the first time on the skin of healthy individuals. Coagulase is considered a virulence factor, hence CoNS are usually regarded as part of the normal human mucous membrane microbiota and are mostly harmless [25]. In addition, the enterobacterium *E. coli,* which does not naturally produce carotenoids, can be genetically modified to express genes from carotenogenic bacteria or plants, generating recombinant strains specifically designed to accumulate the desired carotenoids, including phytoene [26,27].

In this study, methanolic extracts from phytoene-accumulating bacteria and from the microalga *Chlorella sorokiniana*, specially treated to induce the accumulation of phytoene, were characterized. Their potential activity against oxidative damage, their ability to inhibit proliferation and cell migration in several human colonic adenocarcinoma-derived cell lines, and their potential ability to induce the activity of detoxifying antioxidant enzymes, such as NAD(P)H:quinone oxidoreductase and the glutathione S-transferase, were assessed as part of a preliminary study on their protective and antitumor effects.

## 2. Results

### 2.1. Obtaining Functional Extracts from Phytoene-Enriched Algal and Bacterial Biomass

In this study, four functional extracts were obtained from different phytoene-enriched bacterial and microalgal cells. Bacterial extracts were obtained from recombinant *E. coli* bacteria harboring the plasmid pAC-PHYTipi and from the Gram-negative soil bacterium *Sphingomonas echinoides* and the Gram-positive commensal bacterium *Staphylococcus haemolyticus*, pinpointed as good bacterial sources of phytoene by previous screening. The algal extract was obtained from the chlorophyte microalga *Chlorella sorokiniana,* which was chemically induced to accumulate phytoene.

The microalga *C. sorokiniana*, which in normal conditions does not accumulate phytoene, can be treated with the bleaching herbicide norflurazon to block the carotenoid biosynthetic pathway at the level of phytoene desaturase and induce the accumulation of uncolored carotenoids. To determine the optimal non-lethal norflurazon-inducing doses, well-grown cultures (about 1 µg DW biomass mL^−1^) were harvested by centrifugation, suspended in fresh TAP culture medium with growing concentrations of the herbicide norflurazon, between 0.5 and 10 µg L^−1^, and incubated for 48 h. After this incubation, the cultures were harvested by centrifugation at 4400 rpm, and the carotenoids were extracted and analyzed as indicated in Section 4.2 (Appendix A). It was concluded that treatment with 1 μg mL^−1^ of the herbicide for 48 h produced an algal biomass with 6.56 ± 0.13 mg g^−1^ DW of 15Z-phytoene, which represented about 61% of the total carotenoids accumulated by the microalgae. Although the specific growth rate decreased to about 70% of the value observed for the control culture without norflurazon, the microalga retained its growth capacity (Table 1). For higher concentrations of norflurazon, the intracellular concentration of phytoene was lower, as was the specific growth rate and the productivity of total carotenoids, due to the toxic effect of the herbicide.

Regarding the bacteria, *E. coli* harboring the plasmid pAC-PHYTipi, *S. echinoides,* and *S. haemolyticus*, cultured as indicated in Materials and Methods, contained 80.2 ± 12, 306.02 ± 8.63, and 11.03 ± 0.93 μg g^−1^ of 15Z-phytoene, at the end of the exponential phase of growth, respectively (Table 2). The three bacterial strains accumulated phytoene as the only carotenoid: the all-trans phytoene level was undetectable in *E. coli* and *S. haemolyticus*, while it represented around 10% of total phytoene in the case of *S. echinoides* (Appendix A).

Functional extracts were obtained from the indicated algal and bacterial strains, grown (as detailed in Section 4.2) at the optimal conditions for phytoene accumulation, and the content of phytoene and total carotenoids was newly determined in these extracts. The concentration of phytoene in the obtained extracts was 2- and 2.7-fold higher than that in the microbial biomass in the case of *S. echinoides* and *S. haemolyticus*, respectively, and 2.5-fold higher in the case of *E. coli* and the induced microalga *Chlorella sorokiniana* (Table 3).

### 2.2. Protective Effect of the Extracts Against Oxidative Damage

To evaluate the protective ability of the obtained extracts against oxidative damage in T84 CRC cell lines, the cells were subject to pre-treatment for 24 h with two non-toxic doses of each extract and subsequently exposed to oxidative damage by incubation with two doses (1.2 mM and 1.5 mM) of H_2_O_2_.

These experiments revealed that extracts from *E. coli* (pAC-PHYTipi), *S. echinoides,* and *C. sorokiniana* exhibited a statistically significant protection ability against the oxidative damage caused by H_2_O_2_ in the T84 CCRC cell line (Figure 1) (*p* < 0.05). Specifically, *E. coli* (pAC-PHYTipi) extract at 0.01 μg mL^−1^ resulted in 6.6% cellular protection from the oxidative damage caused by 1.5 mM H_2_O_2_ (*p* < 0.05), while *S. echinoides* extract exhibited between 9.3% and 11.9% cell protection across all conditions tested (*p* < 0.01). Furthermore, *C. sorokiniana* at 0.01 μg mL^−1^ resulted in 10% and 7% cellular protection in treatments with 1.2 mM and 1.5 mM H_2_O_2_, respectively (*p* < 0.05). No significant protective activity of the *S. haemolyticus* extract was noted. It was also observed that increasing the extract concentration from 0.01 to 0.1 μg mL^−1^ did not improve protection against oxidative damage.

### 2.3. Effect of the Extracts on the Activity of Detoxifying Enzymes

The possible effect of the different extracts on the activity of two detoxifying enzymes, NAD(P)H:quinone oxidoreductase (QR) and glutathione S-transferase (GST), was tested by exposing HT29 human colonic adenocarcinoma cells to non-toxic doses (0.01 µg mL^−1^) of the extracts for 48 h of treatment (Figure 2). Our results show that exposure to *E. coli* (pAC-PHYTipi), *S. haemolyticus,* and *C. sorokiniana* extracts resulted in a significant increase in QR enzyme activity compared to untreated control cells, increasing this activity by 1.42, 1.3, and 1.2, respectively (*p* < 0.05). For the enzyme GST, interesting results were also obtained, as *E. coli* (pAC-PHYTipi), *S. haemolyticus*, and *S. echinoides* extracts significantly induced the activity of this enzyme. *S. echinoides* extracts showed the highest induction ability, causing an increase in GST activity by 1.34 times compared to the control (*p* < 0.001).

### 2.4. Effect of Extracts on the Migration of T84 and HCT-15 CRC Cells

Cell migration is a key process at the early stages of the complex tumor metastasis process; thus, we studied the effect of *E. coli* (pAC-PHYTipi), *S. echinoides*, *S. haemolyticus,* and *C. sorokiniana* extracts on cell migration (Figure 3).

Cell migratory capacity was measured in the two colon tumor cell lines (T84 and HCT-15) and the control non-tumor L929 murine fibroblast cell line by monitoring wound healing once per day during the assay for a period of 72 h. As shown in Figure 3, the extracts obtained from *S. haemolyticus* and *S. echinoides* triggered a significant decrease in HCT-15 tumor cell migration, with values of 9.3% at 24 h and 9.1% at 72 h, respectively. T84 cancer cell line migration was significantly reduced by *E. coli* (pAC-PHYTipi) extract, by up to 12.4% (*p* < 0.05) and *C. sorokiniana* extract, between 18.7% at 24 h and 8.9% at 72 h of treatment (*p* < 0.01). In contrast, migration of fibroblast L929 control cells was promoted by *S. haemolyticus* and *S. echinoides* by 28.6% and 27.2% at 72 h, respectively (*p* < 0.001).

### 2.5. Effects of Extracts on Proliferation of T84 and HCT-15 Colon Cancer Cell Lines

To evaluate if the four phytoene-rich functional extracts obtained exhibited antiproliferative activity, the T84 and HCT-15 colon cancer cell lines were exposed to increasing concentrations, from 5 to 100 μg mL^−1^, of the indicated extract (Figure 4). The effect of DMSO, used to dissolve the lyophilized extracts, was also analyzed and subtracted from the toxic effect of the corresponding extract. Our results indicate that neither the extracts obtained from *E. coli* nor those obtained from *S. echinoides* showed a significant antitumor effect on the T84 and HCT-15 cell lines below the dose of 100 µg mL ^−1^. Only extracts obtained from *C. sorokiniana* and *S. haemolyticus* at a concentration of 100 µg mL^−1^ exhibited antiproliferative activity in these two cell lines. Extracts from *S. haemolyticus* caused an inhibition of 15% and 20% on the relative proliferation of the T84 and HCT15 cell lines, respectively. Treatment with the extract obtained from the microalga *C. sorokiniana* inhibited the proliferation of T84 and HCT15 cells by 10% and 15%, respectively.

## 3. Discussion

Colorectal cancer is a multifactorial malignant disease with increasing impact in developed countries, whose prevalence has been associated with lifestyle and dietary habits by different epidemiological studies [28]. Although some studies suggest that the intake of vegetables and fruits could be inversely related to colorectal cancer risk, there are contradictory conclusions, and it is still uncertain whether this positive association is true for men and women of all ages and ethnic backgrounds [28,29], or if it can be applied to all kinds of vegetables and fruits [30]. In addition, bacteria in the human microflora can produce bioactive compounds involved in the promotion of health and protection against different pathologies, including cancer. Several studies have demonstrated that alterations in the gut microbiota have a link with tumorigenesis, affecting the immune response or the effectiveness of antitumor agents [31].

Carotenoids are bioactive compounds existing in practically all natural fruits and vegetables, frequently together with other bioactive molecules such as flavonoids, S-compounds, and other bioactive components with well-known antioxidant and anti-inflammatory activities, which in many cases interact with each other or have synergic activities, usually making it a great challenge to investigate the activity of individual compounds [32]. Although the anti-cancer activity of carotenoids, either as isolated compounds or as food ingredients, has been widely studied, the potential activity of phytoene and other colorless carotenoids has been left behind [9]. We obtained extracts of different microalgal and bacterial cells that can accumulate phytoene to investigate their antitumor activity. These studies were performed using whole extracts, due to the importance of studying natural isomeric forms and the importance of the synergic interaction of different molecules present in natural extracts. Moreover, the present tendency in the fight against cancer is not only based on drugs, but also on healthy nutrients. Nutraceutical supplementation may contribute, together with classical chemotherapy, to successful anticancer therapy and can be essential in the response and compliance of patients [33].

In accordance with previous research, we found that treatment of the microalga *Chlorella sorokiniana* with the herbicide norflurazon induced the accumulation of phytoene [22]. After optimizing the herbicide doses, microalgal biomass with high concentrations of phytoene was obtained (Table 1). On the other hand, by the selective expression of relevant genes from carotenoid-producing bacteria in *E. coli*, a transgenic bacterial strain able to accumulate phytoene was obtained (Table 2). This strategy has been successfully used to obtain a large variety of recombinant bacteria able to accumulate different carotenoids [26,27] and has been widely used to carry out functional complementation assays and uncover the function of many genes of the carotenoid biosynthetic pathway [34]. Methanolic extracts of these manipulated microalgal and bacterial strains, in addition to the extracts of two natural bacteria *Sphingomonas echinoides* and *Staphylococcus haemolyticus*, were obtained, and their phytoene content was characterized (Table 3). The antioxidant activity of these extracts and their ability to inhibit proliferation and cell migration in two human colon adenocarcinoma-derived cell lines, line T84 and the drug-resistant line HCT-15, were studied.

Oxidative stress is considered the primary cause of many human health disorders, including cancer [35]. The ability of the methanolic extracts obtained in this study to offer protection against the oxidative damage induced by H_2_O_2_ in the T84 CRC cell line (Figure 1) is in agreement with previous data reported for extracts obtained from other microalgae, such as *Chlorobotrys gloeothece*, *Chlorobotrys regularis,* and *Characiopsis aquilonaris*, which showed high antioxidant activity, analyzed through DPPH assays, correlating in turn in *C. regularis* with a greater amount of β-carotene in breast cancer cell lines [36]. In general, the presence of significant amounts of pigments such as carotenes and chlorophylls, and vitamins, such as vitamin C, have been related to high antioxidant capacity, as demonstrated in *Chlorella vulgaris* [37]. The antioxidant and cell signaling role of carotenoids and their effects in the prevention of cancer have been widely explored and reviewed by several previous reports [38].

In addition, there is increasing evidence that one of the main mechanisms by which many nutritional bioactive components, including carotenoids, contribute to reducing the risk of cancer development is the induction of the activity of detoxifying enzymes, such as glutathione S-transferase (GST) and NAD(P)H:quinone reductase (QR), which play an essential role in the initial detoxification stages of numerous potentially harmful xenobiotics. High levels of these enzymes have been positively correlated with chemoprevention and cancer inhibition. Numerous plant and bacterial extracts containing natural products, such as triterpenoids, vitamins, or carotenoids, have been shown to induce the activity of these enzymes [39,40]. However, the effect of phytoene or phytoene-rich extracts on the activity of GST and QR detoxifying enzymes has not been studied so far. Our results showed that bioactive extracts with a high concentration of phytoene obtained from *E. coli* harboring the pACCRT-EB plasmid, *Staphylococcus haemolyticus*, and *Chlorella sorokiniana*-induced cells resulted in a significant increase in QR enzyme activity compared to untreated control cells. Similarly, *E. coli* (pAC-PHYTipi), *Staphylococcus haemolyticus*, and *Staphylococcus haemolyticus* extracts significantly induced the activity of the GST enzyme (Figure 2).

Finally, the antitumor potential activity of the obtained extracts was evaluated by cytotoxicity and tumor cell migration assays. All the extracts tested in this study showed inhibitory effects on the migration of tumor cells. Interestingly, the best results were obtained in HCT-15, a drug-resistant tumor line, in which *Staphylococcus haemolyticus* and *Sphingomonas echinoides* extracts caused a significant decrease in cell migration by 9.3% at 24 h and 9.1% at 72 h, respectively. In the T84 tumor line, a significant decrease in cell migration was observed upon treatment with *E. coli* (pAC-PHYTipi) and phytoene-enriched *Chlorella sorokiniana* extracts, with decreases of 12.4% and 18.7% after 24 h of treatment (Figure 3). On the other hand, our proliferation results indicated that only the extracts obtained from *C. sorokiniana* and *S. haemolyticus* at a concentration of 100 µg mL ^−1^ exhibited antiproliferative activity in the T84 and HCT15 cell lines. Treatment with the extract obtained from the microalga *C. sorokiniana* inhibited proliferation of these cell lines by 10% and 15%, respectively (Figure 4).

These results are in line with previous studies that explored the antitumor effects of bacterial-derived carotenoids on gastro-intestinal cancer (GIC) cell lines. The studies carried out by Jinendiran and coworkers, who tested the antiproliferative activity of six carotenoid pigments isolated from the bacterial species *Exiguobacterium acetylicum* against HT-29 CRC cells, showed IC_50_ values at 24 h of 115.15, 90.94, 114.88, and 112.52 μM for diapolycopenedioic acid diglycosyl ester, β-carotene, astaxanthin, and keto-myxocoxanthin glucoside ester, respectively. However, lycopene and zeaxanthin were shown to be less effective than other carotenoid pigments in inhibiting the proliferation of these tumor cells [41]. The red carotenoid-related pigments extracted from *Arthrobacter* sp. G20 have also been found to exhibit time-dependent antiproliferative activity against the KYSE-30 cell line, with IC_50_ values of 1321, 668, and 366 μg mL^−1^ at 24, 48, and 72 h, respectively [42]. Analysis of the antitumor activity in vitro of carotenoids obtained from other bacteria such as *Kuria* sp. RAM1 did not show antitumor effects below 125 μg mL^−1^, indicating that it does not affect cell viability, and establishing its IC_50_ at a value of around 500 μg mL^−1^ in breast, colon, and cervix cancer cell lines [43]. Other studies of cellular cytotoxicity in the breast cancer cell line MCF7 and the colon cancer cell line Caco-2 are along the same lines, with results that show that the IC_50_ obtained for carotenoids from *Natrialba* sp. M6 is significantly higher than that obtained in these same cell lines with the antitumor drug 5-fluorouracil [44]. Carotenoid pigments produced by the extremophilic bacteria *Deinococcus* sp. UDEC-P1 and *Arthrobacter* sp. UDEC-A13 have also been evaluated in breast cancer (MCF7), osteosarcoma (Saos-2), and neuroblast (Neuro-2a) cell lines. The pigments obtained in the first case were able to significantly reduce the viability of Saos-2 cells, with a reduction of 37.1%, while no effect was observed against the other cell lines. For the second case, a reduction in cell viability was seen in all the cell lines, in a range from 13 to 26% [45]. Neither of these studies tested phytoene or other uncolored carotenoids. An interesting study carried out by Nishino and coworkers showed the anticarcinogenic activity of several natural carotenoids, including phytoene. Furthermore, these authors reported that the introduction of the crtB gene into mammalian cells yielded phytoene-producing cells which presented higher resistance against carcinogenesis [46].

Bioactive extracts derived from phytoene-producing bacteria and microalgae could play a role in some relevant aspects related to colon cancer, as demonstrated by the results obtained in this study with CRC cancer cell lines. Thus, phytoene-containing microalgae and/or bacteria could be used as a complementary resource in cancer therapy, or as a novel functional food with antioxidant activity to reduce the accumulation of oxidative damage and preventive effect against carcinogenesis. Exploration of the anticancer activity of edible microalgae and bacteria, which can be part of human microbiota, is especially interesting for uncovering new nutraceutical supplements or probiotic microorganisms that can work for the prevention of cancer.

The methanolic extracts used in this study are a complex mixture of many compounds. Although it is not possible to establish an unequivocal relation between the carotenoid content and the observed activities, these results point to these extracts as potential antioxidant complements able to protect cells against oxidative damage and with the ability to inhibit the proliferation and migration of CRC tumor cells, paving the way to design functional foods or probiotic formulations with preventive properties against oxidative stress-related diseases, such as cancer, or as starting point for purifying anticancer compounds. This research opens the possibility of using these extracts as complementary agents in anticancer therapies. In particular, they could be used in combination with conventional treatments, not only for their ability to reduce cell migration and tumor cell proliferation, but also for their potential to mitigate the side effects associated with oxidative stress induced by such treatments.

This study provides a scientific basis to further explore the biological potential of natural extracts enriched with carotenoids, such as phytoene, which have been relatively understudied compared to other carotenoids. Our findings demonstrate that these extracts not only exhibit significant antioxidant activities, but also potential antiproliferative and anti-migratory effects, in colon cancer lines, which could boost the development of new lines of research in the field of oncological pharmacology. This knowledge could be the key for future studies focused on the characterization and synthesis of phytoene derivatives with more specific therapeutic applications.

## 4. Materials and Methods

### 4.1. Microbial Strains and Culture Conditions

Chemically Competent One Shot™ TOP10 *E. coli* cells (Invitrogen, Carlsbad, CA, USA) were transformed with the plasmid pAC-PHYTipi, kindly supplied by Francis X. Cunningham, Jr. (Addgene plasmid # 53283; http://n2t.net/addgene:53283; accessed on 1 July 2024) and cultured in LB medium supplemented with chloramphenicol (30 μg mL^−1^) at 28 °C in the dark. The phytoene-accumulating strains of *Sphingomonas echinoides* (strain *B-3127*) and *Staphylococcus haemolyticus* (CECT 4900) were procured from the ARS Culture Collection NRRL (Department of Agriculture, Washington, DC, USA) and from the Spanish Type culture collection (Valencia, Spain), respectively. *S. echinoides* was cultured in Tryptone–Glucose–Yeast Extract (TGY) medium, pH 7.0, at 26 °C, and *S. haemolyticus* was cultured in nutrient broth/agar II, pH 7.2, at 37 °C. In all cases the cultures were incubated in a rotatory shaker at 150 rpm. The Trebouxiophyceae microalga *Chlorella sorokiniana* (211-32), kindly provided by the algal collection of the Institute of Plant Biochemistry and Photosynthesis (IBVF-CSIC, Seville, Spain), was photomixotrophically cultured in liquid Tris-acetate phosphate (TAP)-enriched medium at 25 °C under continuous white light irradiation (100 µE m^−2^ s^−1^), as previously described [47].

### 4.2. Extract Preparation

Methanol extracts from *C. sorokiniana* and recombinant *E. coli* were obtained by disruption of cells by agitation with glass beads for three cycles of 2 min each at 4 °C with methanol. The extracts from *S. echinoides* and *S. haemolyticus* were obtained by ultrasound-assisted extraction with a 1.6 mm probe (Qsonica, Q500, Newton, CT, USA) at 30% amplitude for 2 min and 20 kHz, using DMSO as solvent. The samples were immersed in an ice bath during sonication to minimize carotenoid degradation. This DMSO extraction process was repeated twice. After the ultrasound extraction, the samples were centrifuged, and the resulting supernatant was transferred to a new tube. Finally, the samples were treated with acetone and centrifuged. All the obtained extracts were concentrated in a rotary vacuum, lyophilized using a freeze drier, and stored at −20 °C under an N_2_ atmosphere. The extracts were re-suspended in either DMSO to study their antitumor activity or ethyl acetate for determination of the carotenoid content.

### 4.3. Analysis of Carotenoids Concentration

Carotenoids were quantified by high-performance liquid chromatography (HPLC) using an Agilent (Palo Alto, Santa Clara, CA, USA) 1260 Infinity II Prime LC System equipped with a diode array detector and a C30 column (3 µm, 150 × 4.6 mm) (YMC, Wilmington, NC, USA). The carotenoid extracts were dissolved in 500 µL of ethyl acetate, and 10 µL was injected into the system for analysis. The wavelengths monitored were 285 nm for the determination of phytoene and 450 nm for the rest of the carotenoids. The mobile phase consisted of a mixture of methanol, tert-butyl methyl ether, and water, which was delivered at a flow rate of 1 mL min^−1^ through a linear gradient, as described in a previous study [48]. The carotenoid content was determined by external calibration, as explained in that study. To determine the total carotenoid content, the sum of all individual carotenoids was calculated. Calibration standards were purchased from Sigma-Aldrich, Inc. (St. Louis, MO, USA) or DHI Lab Products (Hørsholm, Denmark).

### 4.4. In Vitro Culturing and Cell Lines

The T84, HCT-15, and HT29 cell lines were purchased from the American Type Culture Collection (ATCC, Manassas, VA, USA), derived from human colonic adenocarcinoma, as well as L929, a mouse fibroblast cell line. Cells were cultured in Dulbecco’s Modified Eagle’s Medium (DMEM) (Sigma-Aldrich, Madrid, Spain) with supplementation consisting of 10% fetal bovine serum (FBS) and 1% penicillin-streptomycin (Sigma-Aldrich, Madrid, Spain). These cultures were maintained at 37 °C in an atmosphere containing 5% CO_2_.

### 4.5. Cytotoxicity Assessment

The HCT-15 and T84 cell lines were cultured in 48-well microplates, with seeding densities of 6 × 10^3^ and 5 × 10^3^ cells per well, respectively. After an incubation period of 24 h, the cells were subjected to rising concentrations of the bioactive extracts obtained and the solvent (DMSO) over a 72 h interval. The evaluation of cell viability and proliferation was conducted employing a colorimetric assay with sulforhodamine B (SRB) (Sigma-Aldrich, Madrid, Spain) as in Ortiz et al. [49]. Briefly, a 10% solution of trichloroacetic acid (TCA) (Sigma-Aldrich, Madrid, Spain) was applied as a fixative agent at 4 °C for 20 min. Following cell fixation, cells were subjected to staining with a solution comprising 0.4% SRB diluted in 1% acetic acid. Then, the cells were incubated for 20 min at room temperature with gentle agitation. Solubilization of the SRB dye was achieved using Trizma^®^ (Sigma-Aldrich, Madrid, Spain) at a concentration of 10 mM and a pH level of 10.5. The optical density (OD) was quantified at a wavelength of 492 nm using a spectrophotometer EX-Thermo Multiskan (Waltham, MA, USA).

### 4.6. In Vitro Antioxidant Evaluation

T84 cells were cultured in 96-well microplates at a density of 2.5 × 10^4^ cells per well, with each well containing 150 µL of supplemented DMEM. After an incubation period of 24 h, the growth medium was replaced with serum-free DMEM. The following day, treatments using distinct bioactive extracts were administered at two doses that were previously determined as non-toxic to the cells (0.01 and 0.1 μg mL^−1^). These treatments were allowed to interact with the cells for 24 h. Following the treatment period, the culture medium was removed and replaced with fresh serum-free medium. Additionally, certain wells were subjected to treatments with hydrogen peroxide (H_2_O_2_) at concentrations of 1 and 1.5 mM. After an incubation period of 6 h, the medium was once again replaced with fresh serum-free medium and allowed to incubate for an additional 12 h. Assessment of cell viability was conducted by MTT (3-(4,5-Dimethylthiazol-2-yl)-2,5-Diphenyltetrazolium Bromide) assay. In brief, a 10% volume of MTT solution was added to each well and incubated under standard cell culture conditions for 2.5 h. Then, the culture medium was removed, and a mixture comprising 200 μL of DMSO and 25 μL of Sorensen’s glycine buffer (containing 0.1 M glycine, 0.1 M NaCl, pH 10.5 adjusted with 0.1 NaOH) was introduced to dissolve the formazan crystals. The OD of the well contents was measured at a wavelength of 570 nm, with a reference wavelength set at 690 nm, employing the EX-Thermo Multiskan spectrophotometer (Waltham, MA, USA).

### 4.7. Evaluation of Potential Induction of Detoxifying Enzymes

#### 4.7.1. Isolation of Cytosolic Fractions

The isolation of cellular cytosolic fractions was executed using a previous protocol established by our research team [50]. In this context, HT29 cells were seeded in triplicate in six-well plates at an initial cell density of 5 × 10^5^ cells per well. The cells were then incubated for a period of 24 h to facilitate adherence and acclimatization. After this step, the cells underwent exposure to non-cytotoxic concentrations of individual extracts. Additionally, DL-sulforaphane (SFN) (Sigma-Aldrich, Madrid, Spain) at a concentration of 10 µM was included as a positive control. After 48 h, the cells were subjected to a sequence of procedural steps, including washing, trypsinization, and centrifugation. Following two successive washes with phosphate-buffered saline (PBS), the resultant cell pellet was resuspended in 500 µL of Tris-HCl buffer (25 mM, pH 7.4) and subjected to ultrasonic disruption for 20 s, while maintaining the suspension on ice. Subsequently, the cell suspension was centrifuged at 10,000× *g*, and the recovered supernatant, containing the cytosolic fraction, was carefully preserved at a temperature of −80 °C for subsequent analysis. To determine the protein concentration within the isolated cytosolic fraction, Bradford reagent (Bio-Rad, Hercules, CA, USA) was employed.

#### 4.7.2. Measurement of NAD(P)H:Quinone Oxidoreductase Activity

The activity of NAD(P)H:quinone oxidoreductase (QR) was assessed through a colorimetric approach, wherein the reduction of 2,6-dichloroindophenol (DCIP), with a molar extinction coefficient of 0.0205 μmol^−1^ cm^−1^, resulted in a decrease in the OD. For this assay, a reaction mixture was prepared by combining the following components: 881.5 μL of a 25 mM Tris-HCl solution at pH 7.4, 60 μL of 1 mg mL^−1^ bovine serum albumin (BSA) (Sigma-Aldrich, Madrid, Spain), 2.5 μL of 20% Tween, 5 μL of 1 mM flavin adenine dinucleotide disodium (FAD) (Sigma-Aldrich, Madrid, Spain), 10 μL of 20 mM β-nicotinamide adenine dinucleotide (NADH) (Sigma-Aldrich, Madrid, Spain), and 16 μL of 5 mM DCIP. To initiate the reaction, 25 μL of the cytosolic fraction sample was introduced into a plastic cuvette already containing 975 μL of the reaction mixture. The absorbance at 600 nm was recorded at 1 min intervals for a total duration of 5 min. This measurement was performed utilizing a UV-Vis Spectrophotometer UV-1900i (Shimadzu, Duisburg, Germany). Measurement of the QR activity was accomplished by determining the rate of absorbance decreasing per minute, normalized to the milligrams of total protein and compared with untreated cells.

#### 4.7.3. Determination of Glutathione S-Transferase Activity

The enzymatic activity of glutathione S-transferase (GST) was quantified by monitoring the colorimetric transformation resulting from the catalytic reaction facilitated by GST. This reaction involved the interaction between reduced glutathione (GSH) (Sigma-Aldrich, Madrid, Spain) and the GST substrate, 1-chloro-2,4-dinitrobenzene (CDNB) (Sigma-Aldrich, Madrid, Spain). CDNB is known to possess a molar extinction coefficient of 0.0096 μmol^−1^/cm^−1^. Firstly, a reaction mixture was prepared with 870 μL of 100 mM phosphate buffer adjusted to pH 6.5, 20 μL of 50 mM CDNB, and 10 μL of 100 mM GSH. This mixture was incubated at a controlled temperature of 30 °C for 5 min. Then, to start the enzymatic reaction, 100 μL of the cytosolic fraction was introduced into a quartz cuvette already containing 900 μL of the reaction mixture. The absorbance at 340 nm was measured at 1 min intervals for a total duration of 5 min, utilizing a UV-Vis Spectrophotometer UV-1900i (Shimadzu, Duisburg, Germany). The quantification of GST activity was achieved through calculation of the rate of absorbance increase per minute, normalized to the milligrams of total protein, and subsequently compared with untreated cells.

### 4.8. Wound-Healing Assay

A wound-healing assay was conducted using the T84, HCT-15, and L929 cell lines. These cells were cultured in six-well plates, with HCT-15 cells seeded at a density of 4 × 10^5^ cells per well, and T84 and L929 cells at 3 × 10^5^ cells per well. After 24 h, a controlled wound was created in the central region of each well following the established methodology outlined by Grada et al. [51]. The cells were rinsed with PBS, and 1 mL of serum-free DMEM was added to the wells. At the same time, treatment with non-cytotoxic doses of each extract 0.1 μg mL^−1^ was administered. Over the course of the subsequent 72 h, the progress of cell migration to close the wound was systematically monitored at 24 h intervals. Images of the wounds were captured using a DM IL LED microscope (Leica, Wetzlar, Germany). Finally, these images were subjected to quantitative analysis using a dedicated ImageJ “Vessel Analysis”plugin from FIJI 2.9.0 developed by the National Institutes of Health (NIH, Stapleton, NY, USA). The cell-free area within the wound region was measured as previously described by Suarez-Arnedo et al. [52].

### 4.9. Statistical Analysis

The data are presented in the form of mean values accompanied by their respective standard deviations (SDs). For statistical evaluation, we utilized Statistical Package for the Social Sciences (SPSS) software, version 26. Student’s *t*-test was applied, employing a significance level (α) of 5%, to assess the significance of observed differences.

## Figures and Tables

**Figure 1 molecules-29-05003-f001:**
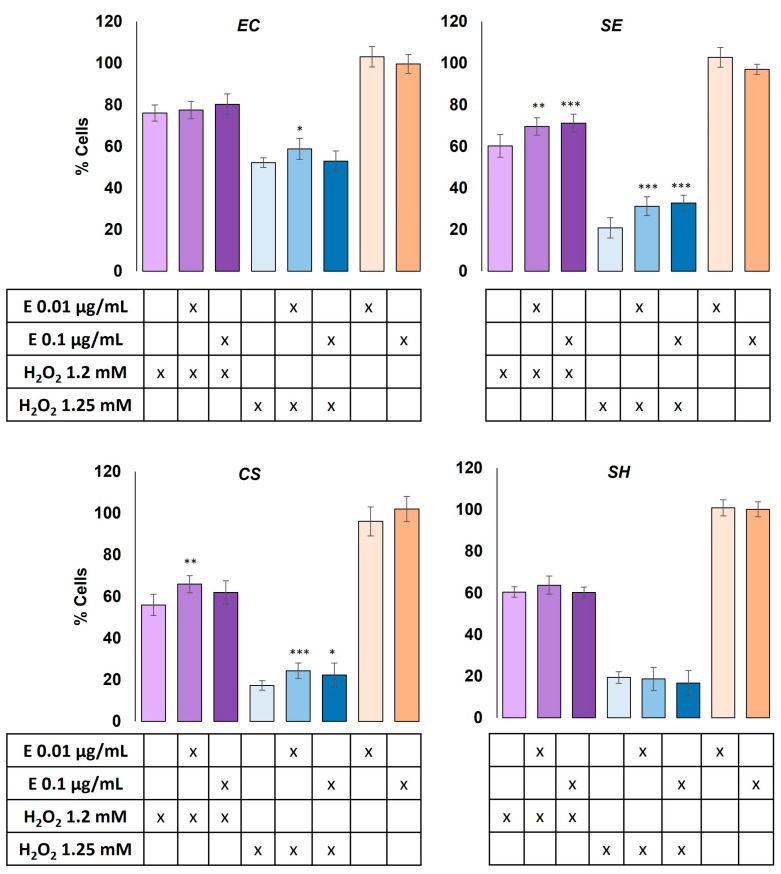
Analysis of the protective potential of methanolic extracts obtained from *E. coli* (EC), *S. echinoides* (SE), *C. sorokininana* (CS), and *S. haemolyticus* (SH). An in vitro antioxidant activity was performed on the T84 cell line pretreated with non-toxic doses (0.01 and 0.1 µg mL^−1^) of each extract and then subjected to oxidative damage by exposure to H_2_O_2_ (1.2 and 1.5 mM). Data are shown as the mean ± SD of eight independent replicates. * *p* < 0.05, ** *p* < 0.01, and *** *p* < 0.001 compared to cells treated with H_2_O_2_ at the doses shown.

**Figure 2 molecules-29-05003-f002:**
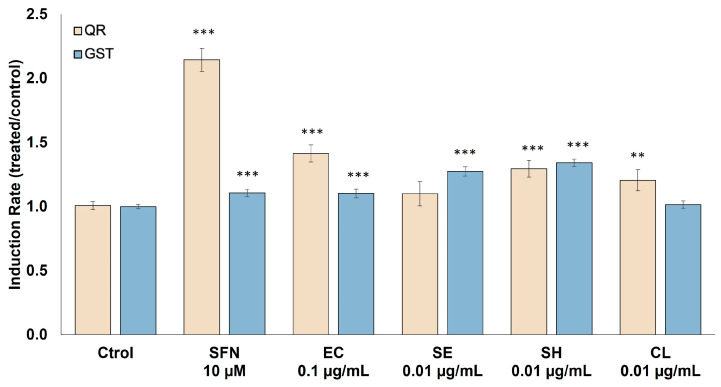
Effect of the extracts on the activity of the detoxifying enzymes QR and GST. After 48 h of treatment with non-toxic doses of each extract (EC: *E. coli* (pAC-PHYTipi); SE: *Sphingomonas echinoides*; SH: *Staphylococcus haemolyticus*; CS: *Chlorella sorokiniana*), the activity of the QR and GST enzymes was determined. DL-sulforaphane (SFN) was established as a positive control. Results are shown as the mean ± standard deviation of three independent experiments. ** *p* < 0.01, and *** *p* < 0.001 compared to untreated cells.

**Figure 3 molecules-29-05003-f003:**
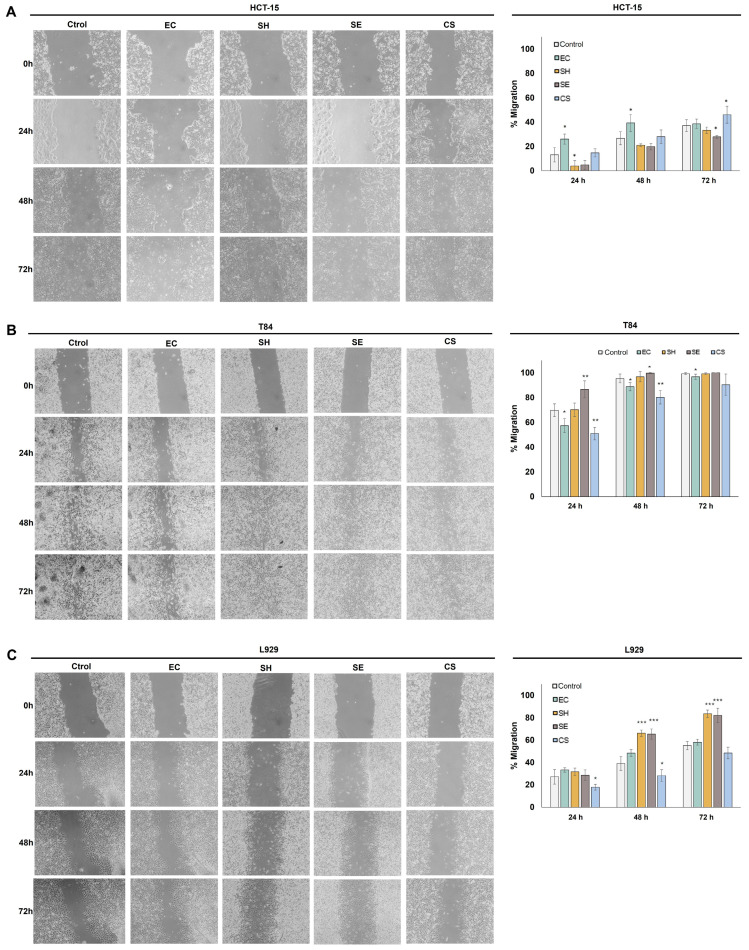
Wound-healing assay carried out to determine the influence of several bioactive extracts (EC: *E. coli* (pAC-PHYTipi); SE: *Sphingomonas echinoides*; SH: *Staphylococcus haemolyticus*; CS: *Chlorella sorokiniana*) on cell migration of two colon cancer cell lines: T84 (**A**) and HCT-15 (**B**), and the non-tumor murine fibroblast cell line L929 (**C**) every 24 h over 72 h of treatment with non-toxic doses of each extract (0.1 μg mL^−1^). Results are expressed as the mean ± standard deviation of at least three independent experiments. * *p* < 0.05, ** *p* < 0.01, and *** *p* < 0.001 compared to control cells.

**Figure 4 molecules-29-05003-f004:**
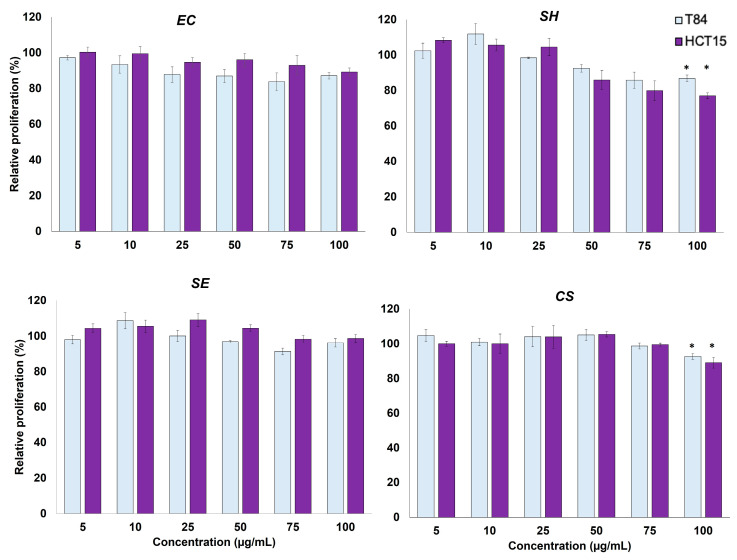
Effect of functional phytoene-rich extracts obtained from the bacteria *E. coli* (pAC-PHYTipi), *S. haemolyticus,* and *S echinoides* and the chlorophyte microalga *C. sorokiniana* on relative proliferation of the colon cancer cell lines T84 and HCT-15 during 72 h of treatment. Data are shown as the mean ± standard deviation of at least three independent experiments. * *p* < 0.05 compared to control cells.

**Table 1 molecules-29-05003-t001:** Assessment of the optimal non-lethal concentration of norflurazon to induce the accumulation of phytoene in *C. sorokiniana* after 48 h of incubation.

Norflurazon (μg mL^−1^)	15Z-Phytoene (mg g^−1^ DW)	Total Carotenoids (mg g^−1^ DW)	Specific Growth Rate µ (h^−1^)
0	0	14.92 ± 0.04	0.055 ± 0.006
0.5	6.45 ± 0.46	10.98 ± 0.77	0.042 ± 0.002
1	6.56 ± 0.13	10.67 ± 1.06	0.038 ± 0.005
1.5	4.97 ± 0.03	10.47 ± 0.21	0.026 ± 0.013
2	0.92 ± 0.49	6.74 ± 0.43	0.023 ± 0.008
5	0.32 ± 0.02	5.56 ± 0.03	0.019 ± 0.003
10	0.1 2 ± 0.02	5.59 ± 0.14	0.015 ± 0.001

Specific growth rate, content of total carotenoids, and phytoene in *C. sorokiniana* cultures treated with increasing concentrations of the herbicide norflurazon. Values are the mean of three independent determinations ± standard deviation. DW: dry weight.

**Table 2 molecules-29-05003-t002:** Intracellular concentration of carotenoids in the bacterial species studied.

Bacterial Species	Culture Conditions	Phytoene (µg g^−1^ DW)	
		(15*Z*)	(All-E)
*E. coli* with pAC-PHYTipi plasmid	LB, Chl (30 μg mL^−1^), 28 °C	80.2 ± 12	n.d.
*Sphingomonas echinoides*	TGY, 26 °C	11.03 ± 0.93	n.d.
*Staphylococcus haemolyticus*	NB, 37 °C	306.02 ± 8.63	30.17 ± 1.19

Content of phytoene in three different bacterial strains, cultured at the indicated conditions. Values are the mean of three independent determinations ± standard deviation. DW: dry weight. Chl: chloramphenicol. n.d.: not detected.

**Table 3 molecules-29-05003-t003:** Concentration of carotenoids in the functional extracts obtained.

Species	15*Z*-Phytoene (µg g^−1^)	All-*E*-Phytoene (µg g^−1^)	Total Carotenoids (µg g^−1^)
Phytoene-enriched *E. coli*	193 ± 28	n.d.	193 ± 28
*Sphingomonas echinoides*	22.06	n.d.	22.06
*Staphylococcus haemolyticus*	765.05	75.43	840.48
Phytoene-enriched *Chlorella*	10,335 ± 90	n.d.	16,670 ± 180

Content of phytoene isomers and total carotenoids (TCCs) in four functional extracts obtained from the indicated species. Values are the mean of three independent determinations ± standard deviation. n.d.: not detected; DW: dry weight.

## Data Availability

Data are contained within the article.

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
