# Peer review of "Preliminary Assessment of the Protective and Antitumor Effects of Several Phytoene-Containing Bacterial and Microalgal Extracts in Colorectal Cancer"

_molecules, 2024, doi:10.3390/molecules29215003_

Round 1
Reviewer 1 Report
Comments and Suggestions for Authors
It is an excellent work, very well organized and very well presented .
Comments on the Quality of English LanguageMinor mistakes on words eg letters missing should be adressed
For instance see lines 20, 323, 389.
Author Response
ANSWERS TO REVIEWER 1:
1.Comments and Suggestions for Authors. It is an excellent work, very well organized and very well presented.
Thank you very much for your rating, we really appreciate it.
- Comments on the Quality of English Language. Minor mistakes on words eg letters missing should be addressed. For instance, see lines 20, 323, 389.
Thank you for your appreciation, the errors have been corrected.
Reviewer 2 Report
Comments and Suggestions for Authors
The paper “Preliminary assessment of the protective and antitumor effects of several phytoene-containing bacterial and microalgal extracts in colorectal cancer” presented the characteristics and potential activity against oxidative damage and the ability to inhibit proliferation and cell migration in several human colon-adenocarcinoma-derived cell lines of several bacterial and algal phytoene-enriched extracts. The research is abundant in content and possesses certain levels of innovation and practical applicability. However, the research scope is excessively broad, leading to a lack of depth. Below are relevant specific issues.
Comments:
Q1. The full name of CRC should be marked for the first time in the main text (line 37).
Q2. The rationale behind the author's decision to include the structural representation section (some Figures) in the supplementary document remains unclear; if this is indeed the case, it results in a significant deficiency of data within the main text.
Q3. Based on the available data, 90% of the components in the extract are uncertain, as 10mg/g equals 1%. In this case, it is impossible to determine what specific substances in the article are responsible for its effects. The author is advised to supplement data related to the structure.
Q4. What is the rationale for utilizing tumor cells instead of normal cells in the development of a model for oxidative damage? Is there a necessity to provide protection to tumor cells?
Q5. The “u” should be changed to “μ”, please check these figures.
Q6. There are some mislabeled “.” and blank spaces in the text. Please check the entire text carefully.
Q7. What criteria were used to determine the dosage in this study, and what were the findings of the preliminary experiments? Additionally, given the low dosage and the uncertainty regarding the active ingredients, the data exhibit a limited degree of reproducibility.
Q8. The data from the cell experiments presented in this paper are insufficient for acceptance; therefore, it is strongly recommended that the authors provide additional information regarding the extraction and structural characterization of the compounds in the first half of the manuscript. Furthermore, they should elucidate the structural differences among various extracts and their underlying causes, while correlating these findings with the cellular activity reported later in the paper.
Comments on the Quality of English LanguageThere are some mislabeled “.” and blank spaces in the text. Please check the entire text carefully. Besides, some grammatical errors also need to be carefully checked.
Author Response
ANSWERS TO REVIEWER 2:
Comments and Suggestions for Authors. The paper “Preliminary assessment of the protective and antitumor effects of several phytoene-containing bacterial and microalgal extracts in colorectal cancer” presented the characteristics and potential activity against oxidative damage and the ability to inhibit proliferation and cell migration in several human colon-adenocarcinoma-derived cell lines of several bacterial and algal phytoene-enriched extracts. The research is abundant in content and possesses certain levels of innovation and practical applicability. However, the research scope is excessively broad, leading to a lack of depth. Below are relevant specific issues.
Comments:
Q1. The full name of CRC should be marked for the first time in the main text (line 37).
The full definition of CRC as Colorectal Cancer has been added on line 37
Q2. The rationale behind the author's decision to include the structural representation section (some Figures) in the supplementary document remains unclear; if this is indeed the case, it results in a significant deficiency of data within the main text.
The main objective of this study is analyze the effect of several microbial extracts on: i) in vivo antioxidant activity of the extracts (Fig. 1); ii) Induction of detoxifying enzymes (QR, GST) (Fig. 2); iii) Cell migratory capacity (Fig. 3) and iv) Proliferation of CRC cell lines (Fig. 4). The chromatographic analysis of the carotenoid composition of the extracts has been carried out to confirm the presence of phytoene in the selected biomass, and data have been showed in Tables 1-3. The chromatographic analysis of the extracts has been included as five supplementary figures, to avoid redundancies and an excessive number of figure. Although they could also be shown as standard figures if required.
Q3. Based on the available data, 90% of the components in the extract are uncertain, as 10 mg/g equals 1%. In this case, it is impossible to determine what specific substances in the article are responsible for its effects. The author is advised to supplement data related to the structure.
The main aim of this study is to assess the potential activity against oxidative damage and the ability to inhibit the proliferation and migration of CRC tumor cells of different total natural microbial extracts. The study of whole extracts is the first step in the procedure to discover new potentially interesting nutraceuticals and it is the usual approach used, as shown in the literature, for extracts from diverse origins (Ajmal et al., 2022), including microorganisms (Assunção et al 2017) or plants (Rodrigues et al., 2016, Saeed and Shabbir 2012). These are only some examples of the big amount of papers and reviews that can be found in the literature that study whole extracts as a first strategy to uncover the bioactivity potential of plants and microorganisms components.
Of course, as suggested by referee 2, further studies focused on the identification and isolation of the molecule or molecules responsible for these effects, the determination of the quantities and the proportions of each one in the extract or the mechanism of action are desirable. Aa a matter of fact, on the basis of this preliminary assessment, the bio-guided fractionation of these extracts have been programed, however this is not the objective of this study. Comments about these limitations and perspectives have been added to the discussion of manuscript.
- Ajmal et al (2022). Evaluation of Anti-cancer and Anti-proliferative Activity of Medicinal Plant Extracts (Saffron, Green Tea, Clove, Fenugreek) on Toll Like Receptors Pathway Natural Product Sciences. 28(3): 121-129 (2022). https://doi.org/10.20307/nps.2022.28.3.121
- Rodrigues, M. J., Neves, V., Martins, A., Rauter, A. P., Neng, N. R., Nogueira, J. M., ... & Custódio, L. (2016). In vitro antioxidant and anti-inflammatory properties of Limonium algarvense flowers’ infusions and decoctions: A comparison with green tea (Camellia sinensis). Food Chemistry, 200, 322-329. https://doi.org/10.1016/j.foodchem.2016.01.048
- Assunção, M.F.G., Amaral, R., Martins, C.B. et al.Screening microalgae as potential sources of antioxidants. J Appl Phycol 29, 865–877 (2017). https://doi.org/10.1007/s10811-016-0980-7
- Saeed, N., Khan, M.R. & Shabbir, M. Antioxidant activity, total phenolic and total flavonoid contents of whole plant extracts Torilis leptophyllaBMC Complement Altern Med 12, 221 (2012). https://doi.org/10.1186/1472-6882-12-221
Q4. What is the rationale for utilizing tumor cells instead of normal cells in the development of a model for oxidative damage? Is there a necessity to provide protection to tumor cells?
We appreciate your comment and understand your concerns about the relevance of performing an antioxidant capacity assay in a tumor cell line. However, we consider this assay appropriate and relevant in the context of this study for the following reasons.
Oxidative stress plays a fundamental role in the development and progression of cancer (Reuter et al., 2010). Reactive oxygen species (ROS), such as hydrogen peroxide, can promote DNA damage, mutations, and alterations in cell signaling pathways that favor tumor proliferation. Therefore, studying antioxidant capacity in tumor cells and their modulation by antioxidant agents or specific therapies is relevant to understand how cancer cells handle this oxidative stress.
In addition, cancer cells have an altered metabolism and tend to have higher ROS levels than normal cells (Gorrini et al., 2013). Evaluating antioxidant capacity in these cells can provide valuable information on how tumor lines respond to different treatments, or on how certain compounds can influence the redox balance within cancer cells, affecting their survival. Previous studies, as Trachootham et al., have shown that manipulating ROS levels and antioxidant capacity in cancer cells can influence their proliferation and susceptibility to treatments such as chemotherapy and radiotherapy (2009). Therefore, assessing antioxidant capacity in a tumor cell line could provide relevant information in the context of therapeutic strategies that seek to alter the redox balance of tumor cells.
Gorrini, C.; Harris, I.S.; Mak, T.W. Modulation of Oxidative Stress as an Anticancer.
Strategy. Nat Rev Drug Discov 2013, 12, 931–947, doi:10.1038/nrd4002.
Reuter, S.; Gupta, S.C.; Chaturvedi, M.M.; Aggarwal, B.B. Oxidative Stress, Inflammation, and Cancer: How Are They Linked? Free Radic Biol Med 2010, 49, 1603–1616, doi:10.1016/j.freeradbiomed.2010.09.006.
Trachootham, D.; Lu, W.; Ogasawara, M.A.; Valle, N.R.-D.; Huang, P. Redox Regulation of Cell Survival. Antioxid Redox Signal 2008, 10, 1343–1374, doi:10.1089/ars.2007.1957.
Q5. The “u” should be changed to “μ”, please check these figures.
Thank you very much for your comment, “u” has been substituted by “μ” in figure 1 and through the text.
Q6. There are some mislabeled “.” and blank spaces in the text. Please check the entire text carefully.
Thank you very much for your comment. The text has been completely revised, with emphasis on blank spaces and punctuation.
Q7. What criteria were used to determine the dosage in this study, and ? Additionally, given the low dosage and the uncertainty regarding the active ingredients, the data exhibit a limited degree of reproducibility.
Thank you for your question regarding the selection of the doses used in our tests. For the cell toxicity test, concentrations between 5 and 100 µg/mL were selected. These doses were chosen considering several factors. Firstly, the toxicity of the solvent used, DMSO, was considered, which can present toxic effects at higher concentrations. For this reason, higher doses were limited to avoid interference. In addition, the concentrations used were also determined by the initial concentration of the sample, thus ensuring that the tests were representative and relevant in terms of their application.
For the antioxidant capacity and cell migration assay non-toxic doses previously established in the scientific literature were used. We based our work on the concentrations reported in our previous article “Extracts from Microalgae and Archaea from the Andalusian Coast: A Potential Source of Antiproliferative, Antioxidant, and Preventive Compounds”, published in the Journal of Marine Science and Engineering. In this study, similar extracts were investigated.
Luque, C.; Perazzoli, G.; Gómez-Villegas, P.; Vigara, J.; Martínez, R.; García-Beltrán, A.; Porres, J.M.; Prados, J.; León, R.; Melguizo, C. Extracts from Microalgae and Archaea from the Andalusian Coast: A Potential Source of Antiproliferative, Antioxidant, and Preventive Compounds. Journal of Marine Science and Engineering 2024, 12, 996, doi:10.3390/jmse12060996.
We also appreciate your comments regarding the reproducibility of the data and the relationship to the doses used and the uncertainty about the active ingredients. The concentrations used in our trials, although they may seem low, were carefully selected based on previous studies that demonstrated the effectiveness of similar extracts. Furthermore, as detailed in our methodology, these doses were chosen to avoid toxic effects and to ensure that the observed responses were not the result of general toxicity, but rather of the specific biological activity of the extract. Similar doses have been used and validated in previous studies, which supports the applicability of our concentrations for the effects evaluated.
We are aware that natural extracts, such as those used in this study, may have multiple active components, which can generate some variability in the results. However, we have made a significant effort to standardize the experimental conditions to minimize such variability. Furthermore, our work aims to identify the bioactive potential of the extracts, and the results obtained provide a solid basis for future, more specific studies that could focus on the characterization of the main fractions of the extract with activity and eventually discover the active compounds.
Q8. The data from the cell experiments presented in this paper are insufficient for acceptance; therefore, it is strongly recommended that the authors provide additional information regarding the extraction and structural characterization of the compounds in the first half of the manuscript. Furthermore, they should elucidate the structural differences among various extracts and their underlying causes, while correlating these findings with the cellular activity reported later in the paper.
This study provides important information to other researchers trying to find bioactive compounds against these tumor cell lines, particularly the drug-resistance CRC cell line HCT-15.
These studies have been performed using whole extracts due to the importance of studying natural isomeric forms and the importance of the synergic interaction of molecules that are only present in natural extracts. Moreover, the present tendency in the fight against cancer is not only based on drugs, but also in healthy nutrients. Nutraceucetical supplementation may contribute, together with classical chemotherapy to successful anticancer therapy and can be essential in the response and compliance in patients. At this stage of our research it is more important elucidate the bioactivity of the whole extracts to identify the best candidates for the next studies, than the detailed characterization of all the compounds present in the extracts
It is noticeable that despite the low concentration at which the potentially bioactive compounds are present in the natural extracts, it is possible to observe activity against the tumour cells. In this study we have forced the accumulation of the carotenoid phytoene in microalgae and we have selected bacteria able to accumulate it, but of course it cannot be confirmed that this molecule is the direct responsible of the activities of the extracts. This has been more clearly stated in the manuscript. Exploration of the anticancer activity of edible microalgae and of bacteria which can be part of human microbiota is especially interesting for uncovering nutraceutical supplements or probiotic microorganisms that can work for prevention of cancer
Our next studies will focus in a bio-guided fractionation of these extracts trying to identify the best fractions and address their characterization. This is the usual pipeline when working with natural extracts. There are hundreds of publications with this approach, which differs greatly from the one used when the bioactivity of a synthetic compound is being tested. In these cases, it is common to carry out a detailed characterization of the molecules present after each reaction.
Following the indications of referee 2, all this has been clarified in the manuscript
Reviewer 3 Report
Comments and Suggestions for Authors
1. Could the Authors provide correlation analyses of the analyses performed? In particular what are the correlations between the analyses of the tests and the quantities of carotenoids extracted?
2. Could the Authors more thoroughly discuss the practical perspectives of the use of their findings and draw some conclusions in relation to the potential use of their results?
Author Response
ANSWERS TO REVIEWER 3:
Comments and Suggestions for Authors
- Could the Authors provide correlation analyses of the analyses performed? In particular what are the correlations between the analyses of the tests and the quantities of carotenoids extracted?
Given the variety of compounds present in the whole extracts it is not possible to stablish a univocal relation between the carotenoid content and the regarded bioactivity. It is possible to stablish correlations between the quantity of extract and activity tested, as has been done in the case of the in vivo antioxidant activities (Fig.1) and the antiproliferative activity over the two tumor lines studied (Fig. 4).
The in vivo antioxidant activity against (Fig. 1) was determined at two concentrations of the extracts (0.1 and 0.01 uµ/ml) and it was observed that increasing the extract concentration from 0.01 to 0.1 µg/ml did not improve the protection against the oxidative damage, indicating that the low doses was enough to saturate the antioxidant activity, this has been indicated in the manuscript.
Regarding the antiproliferative activity over the two tumor lines studied (Fig. 4), the effect of the extracts was only significant for the SH and CS extracts at the highest concentration tested, so no dose correlation can be stablished. In this case the best response was for SH and CS extracts, which are also those with the highest concentration of phytoene. In cases of detoxifying enzymes (Fig. 2) and cell migratory capacity (Fig. 3) the extracts were tested at one unique concentration so it is not possible to stablish a dose correlation.
These has been more clearly explained in the new version of manuscript
- Could the Authors more thoroughly discuss the practical perspectives of the use of their findings and draw some conclusions in relation to the potential use of their results?
Thank you very much for your comment on the need to further elaborate on practical perspectives and conclusions regarding the potential use of our results. A new paragraph has been included at the end of the conclusion section, discussing these issues.
“.these results point to these extracts as potential antioxidant complements able to protect cells against oxidative damage and with ability to inhibit the proliferation and migration of CRC tumor cells, paving the way to design functional foods or probiotic formulations with preventive properties against oxidative stress-related diseases, such as cancer, or as starting point for purifying anticancer compounds. This opens the possibility of using these extracts as complementary agents in anticancer therapies. In particular, their use in combination with conventional treatments, not only for their ability to reduce cell migration and tumor cell proliferation, but also for their potential to mitigate side effects associated with oxidative stress induced by such treatments.
This study provides a scientific basis to further explore the biological potential of natural extracts enriched with carotenoids, such as phytoene, which have been relatively understudied compared to other carotenoids. Our findings demonstrate that these ex-tracts not only exhibit significant antioxidant activities, but also potential antiproliferative and antimigratory effects in colorectal cancer lines, which could boost the development of new lines of research in the field of oncological pharmacology. This knowledge could be the key for future studies focused on the characterization and synthesis of derivatives obtained from these extracts with more specific therapeutic applications”.
Round 2
Reviewer 2 Report
Comments and Suggestions for Authors
Thanks for the author's revision and explanation. The only concern is that the uncertainty of chemical composition will reduce the reproducibility of the results of this paper by other scholars.